# TAF7L modulates brown adipose tissue formation

Haiying Zhou[1,2], Bo Wan[3,4], Ivan Grubisic[1,2], Tommy Kaplan[5], Robert Tjian[1,2]*

[1]Department of Molecular and Cell Biology, Howard Hughes Medical Institute, University of California, Berkeley, Berkeley, United States; [2]CIRM Center of Excellence, Li Ka Shing Center For Biomedical and Health Sciences, University of California, Berkeley, United States; [3]State Key Laboratory of Genetic Engineering, School of Life Sciences, Fudan University, Shanghai, China; [4]California Institute of Quantitative Biosciences, University of California, Berkeley, Berkeley, United States; [5]School of Computer Science and Engineering, The Hebrew University of Jerusalem, Jerusalem, Israel

**Abstract** Brown adipose tissue (BAT) plays an essential role in metabolic homeostasis by dissipating energy via thermogenesis through uncoupling protein 1 (UCP1). Previously, we reported that the TATA-binding protein associated factor 7L (TAF7L) is an important regulator of white adipose tissue (WAT) differentiation. In this study, we show that TAF7L also serves as a molecular switch between brown fat and muscle lineages in vivo and in vitro. In adipose tissue, TAF7L-containing TFIID complexes associate with PPARγ to mediate DNA looping between distal enhancers and core promoter elements. Our findings suggest that the presence of the tissue-specific TAF7L subunit in TFIID functions to promote long-range chromatin interactions during BAT lineage specification.

*For correspondence: tjianr@hhmi.org

Recent studies found that adult humans retain active BAT depots capable of sustaining elevated basal metabolic rates compared to WAT and those higher levels of BAT correlate with lower body mass index (*Carey and Kingwell, 2013*; *Harms and Seale, 2013*; *Chechi et al., 2014*; *Kajimura and Saito, 2014*). These findings inspired heightened efforts to better understand the formation of BAT during mammalian development. Previously, we showed that TAF7L, an orphan TBP-associated factor cooperates with PPARγ in directing WAT gene regulation (*Zhou et al., 2013a*). Given the central role of PPARγ in both WAT and BAT development (*Cannon and Nedergaard, 2004*; *Kajimura et al., 2010*), we wondered whether TAF7L might also serve as a co-activator regulating the formation of BAT.

To test this hypothesis and assess the functional requirement for TAF7L during BAT development, we performed haematoxylin & eosin (H&E) staining and immunostaining using FABP4 and UCP1 antibodies to delineate regions of BAT in wild-type (WT) and *Taf7l* knockout (KO) embryos. As expected, FABP4 stains both WAT and BAT, while UCP1 only stains BAT (*Pedersen et al., 2001*), which is filled with multilocular lipid droplets (*Figure 1A*). We found that UCP1[+] BAT in *Taf7l* KO mice become largely disorganized, is significantly reduced in size and contains decreased lipid levels (*Figure 1A*, *Figure 1—figure supplement 1A–C*). Measurement of dissected BAT pads showed a ~40% weight reduction of BAT in *Taf7l* KO animals compared to WT embryos (*Figure 1—figure supplement 1B*). Intriguingly, skeletal muscle-like tissue emerges and invades regions that normally contain BAT in *Taf7l* KO mice (*Figure 1A*, *Figure 1—figure supplement 1D*), as revealed by a more pronounced red color of *Taf7l* KO BAT and the staining of pan-skeletal muscle marker myosin heavy chain (MYHC) (*Beylkin et al., 2006*; *Figure 1—figure supplement 1C,E*). Our data suggest that loss of TAF7L substantially alters the relative proportion of BAT and muscle lineages in vivo.

Next, we analyzed global gene expression profiles in WT and *Taf7l* KO BAT by RNA-sequencing (mRNA-seq) (*Figure 1B*). Over a thousand genes exhibited altered expression levels by >threefold upon *Taf7l* KO. In particular, BAT-selective genes such as *Ucp1*, *Pgc1α*, *Cidea*, and *Scd1* involved in

**eLife digest** Mammals produce two distinct types of adipose tissue: white adipose tissue (white fat) is the more common type and is used to store energy; brown adipose tissue (brown fat) is mostly found in young animals and infants, and it plays an important role in dissipating energy as heat rather than storing it in fat for future use. In adults, higher levels of brown fat are associated with lower levels of fat overall, so there is considerable interest in learning more about this form of fat to help address rising levels of obesity in the world.

Building on previous work in which they showed that a gene control protein called TAF7L has a central role in the development of the cells that make up white adipose tissue, Zhou et al. now show that this protein also helps to regulate the development of brown adipose tissue.

Mice lacking the gene for this protein developed embryos with 40% less brown fat than wild-type mice with the gene. Moreover, these mice developed muscle-like cells in the regions that should have contained brown fat. Gene expression analysis revealed that 'knocking out' the gene for TAF7L changed the expression of more than a thousand genes in these mice.

Zhou et al. suggest that TAF7L works as a 'molecular switch' that determines whether certain precursor cells (called mesenchymal stem cells) go on to become brown fat cells or muscle cells. A future challenge will be to devise interventions to regulate the activity or levels of TAF7L as a potential means of modulating brown fat depots in animals and humans.

brown fat differentiation, thermogenesis, and mitochondrial function became significantly down-regulated (*Figure 1C*, *Figure 1—figure supplement 2*; *Harms and Seale, 2013*; *Ohno et al., 2013*). Consistent with the observed enhanced skeletal muscle morphological phenotype (*Figure 1A*), we observed a concomitant up-regulation of skeletal muscle genes including *Myh1-8*, *Myf5*, and *Pax3* (*Figure 1C*, *Figure 1—figure supplement 2*; *Abe et al., 2013*). Loss of TAF7L also led to activation of select genes in the formation of cartilage and bone development, but the majority of up-regulated genes appear to be involved in skeletal muscle development and function. We speculate that because BAT and muscle share common MYF5[+] dermotomal precursors (*Kajimura et al., 2009*; *Seale and Lazar, 2009*; *Seale et al., 2011*), this relationship might favor the switch from BAT to skeletal muscle as shown in *Figure 1A*. Our findings suggest that TAF7L tips the balance in favor of BAT development at the expense of skeletal muscle.

To examine the effects of TAF7L loss on brown adipocyte differentiation in vitro, we used C3H10T1/2 mesenchymal stem cells which is able to form multiple cell lineages including adipocytes, muscle, cartilage, and bone. We first depleted TAF7L levels in C3H10T1/2 cells by RNA interference using previously described shTAF7L and control shGFP constructs followed by induction of BAT differentiation (*Zhou et al., 2013a*). As expected, we observed efficient formation of round fat cells peaking at day 2 post-induction in C3H10T1/2 cells treated with shGFP controls (*Figure 2A*). By contrast, TAF7L-depleted C3H10T1/2 cells formed elongated muscle-like cells rather than round fat laden adipocytes (*Figure 2A*). mRNA analysis of post-induction cells (day2) revealed that specific muscle markers (*Myf5*, *Myod1*, and *Mef2c*) become activated in cells treated with shTAF7L, in some cases reaching up to ~30% of their expression level in myotubes (*Figure 2B*). At the same time, protein (PPARγ and UCP1) and mRNA (*Pgc1α*, *Ucp1*, and *Cidea*) levels of brown adipocyte markers become significantly reduced 5 days post-induction, suggesting an efficient blockade of brown fat differentiation (*Figure 2C,D*). We confirmed these results using primary WT and *Taf7l* KO brown adipocytes (*Figure 2—figure supplement 1*). Our findings suggest that loss of TAF7L shifts mesenchymal stem cells in culture from adopting an adipocyte fate toward formation of muscle-like cells.

We next examined TAF7L gain-of-function in vitro using C2C12 myoblasts that can efficiently form myotubes in vitro. Instead of treating C2C12 cells with the normal muscle-inducing protocol, we applied the brown fat differentiation regime to either control C2C12 cells (C2C12.CNTL) or myoblasts ectopically expressing TAF7L (C2C12.TAF7L) and found that only C2C12.TAF7L cells form multilocular brown fat cells (*Figure 2E*; *Zhou et al., 2013a*). RNA analysis showed that TAF7L in C2C12 cells represses the expression of myoblast genes *Myf5* and *Myod*1 while significantly increasing the expression of brown fat-selective genes (*Ucp1*, *Cidea*, *Pgc1α,* and *Pparα)* post-differentiation (*Figure 2F,G*, *Figure 2—figure supplement 2*), suggesting that TAF7L can drive myoblasts toward the brown fat

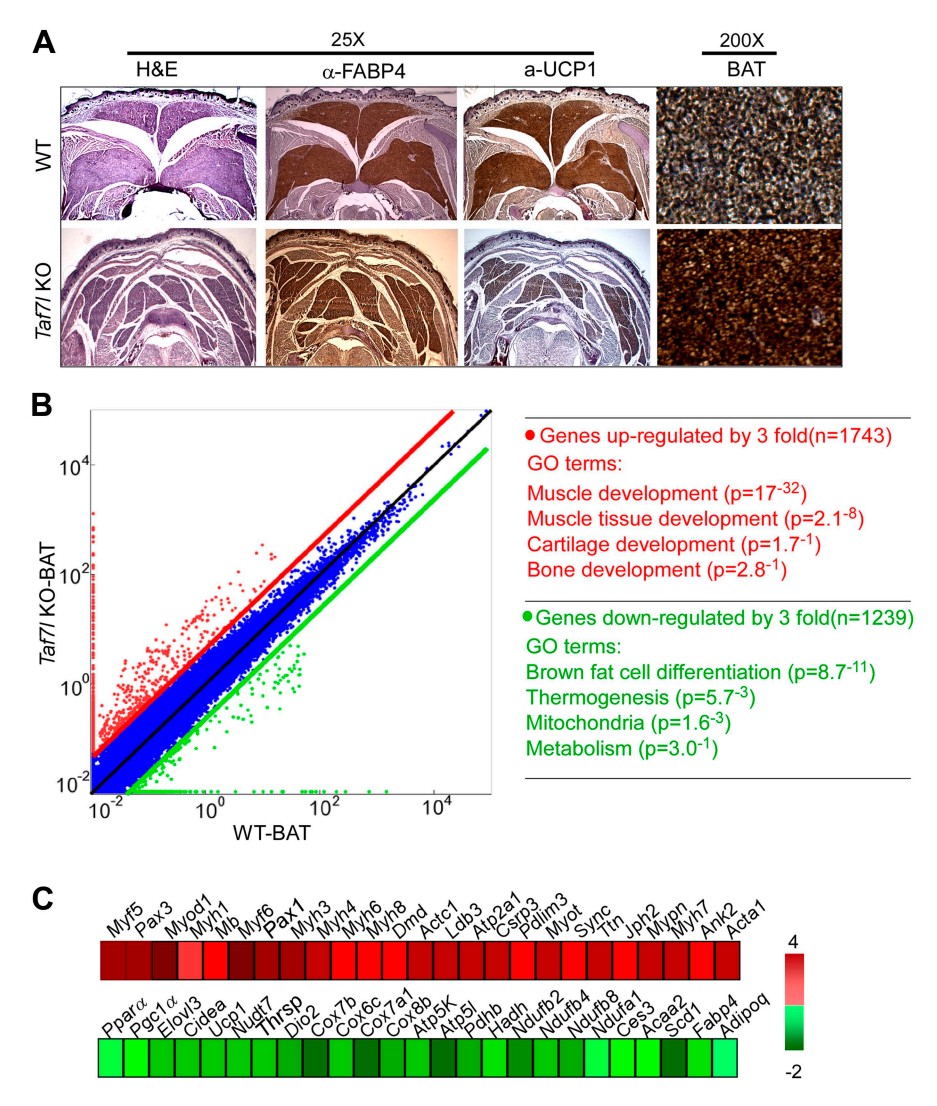

**Figure 1**. TAF7L is required for proper mouse brown adipose tissue (BAT) formation. (**A**) Wild type (WT) and *Taf7l* knockout (*Taf7l* KO) interscapular BAT from E18.5 embryos was stained with haematoxylin and eosin (H&E), FABP4, and UCP1 (25X magnification shown). Rightmost panel shows FABP4 staining at 200X magnification. (**B**) Left panel: scatterplot shows gene expression profile in WT versus *Taf7l* KO BAT tissue; red dots represent genes up-regulated in *Taf7l* KO, green dots represent down-regulated genes in *Taf7l* KO. Right panel: gene ontology analysis shows functional groups of genes changed by at least threefold. (**C**) Difference in expression of either muscle-specific (top) or brown fat (bottom) genes between WT and *Taf7l* KO BAT with progressive red and green shades showing degrees of up- and down-regulation, respectively.

The following figure supplements are available for figure 1:

**Figure supplement 1**. Depletion of TAF7L shifts BAT to muscle lineage.

**Figure supplement 2**. RT-qPCR comparison of gene expression in WT and *Taf7l* KO BAT.

lineage (*Seale et al., 2008*). These gain of function in vitro results are consistent with the switch from BAT to muscle we observed in *Taf7l* KO animals in vivo.

Previously, we showed that TAF7L associates with PPARγ when overexpressed in 293T cells (*Zhou et al., 2013a*). Here, we wanted to confirm this interaction in C3H10T1/2 cells upon brown fat induction. By using sequential immunoprecipitations of doubly tagged TAF7L (FLAG&V5), followed by

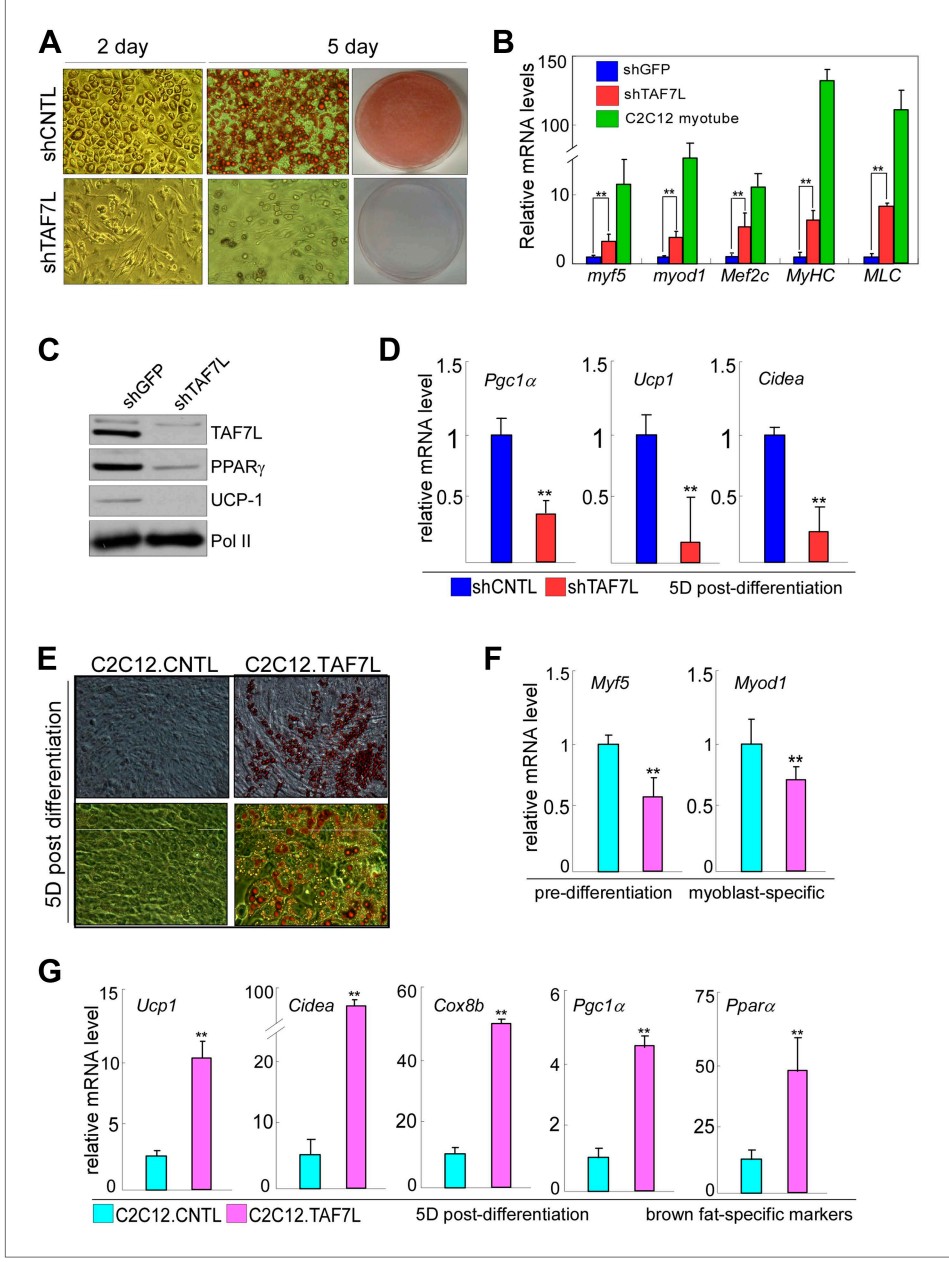

**Figure 2**. TAF7L bidirectionally regulates BAT/muscle lineages. (**A**) 2 days (2D) and 5 days (5D) post-induced control (shCNTL) and TAF7L knockdown (shTAF7L) C3H10T1/2 cells stained with Oil Red O. (**B**) mRNA levels of myoblast genes *Myf5*, *Myod1*, *Mef2c*, *Myhc*, and *Mlc* in 2D post-induced shCNTL and shTAF7L C3H10T1/2 cells, compared to C2C12-induced myotubes. (**C**) Protein levels of TAF7L, PPARγ, and UCP1 in 5D post-induced shCNTL and shTAF7L cells; Pol II was used as loading control. (**D**) mRNA levels of brown fat genes (*Pgc1a*, *Ucp1*, and *Cidea*) in 5D post-induced control and shTAF7L cells. (**E**) 5D post-induced control (C2C12.CNTL) and TAF7L-expressing C2C12 (C2C12.TAF7L) cells were stained with Oil Red O. (**F**) mRNA levels of myoblast genes *Myf5* and *Myod1* in pre-induction cells. (**G**) mRNA levels of BAT-specific genes (*Ucp1*, *Cidea*, *Cox8b*, *Pgc1a*, and *Pparα*) 5D post-differentiation. *$p < 0.05$, data is mean and SEM is from triplicates.

The following figure supplements are available for figure 2:

**Figure supplement 1**. TAF7L is required for brown fat cell differentiation from primary brown adipocytes.

**Figure supplement 2**. Gene expression analysis of control (CNTL) and TAF7L-expressing (TAF7L) C2C12 cells from 0 (0D) to 5 days (5D) post BAT-induction.

liquid chromatography coupled with tandem mass spectrometry (LC-MS/MS) and western blotting, we found that TAF7L pulls down a subset of TAFs normally associated with the prototypical TFIID complex. Although this coIPed complex contained a number of canonical TFIID TAF subunits, we nevertheless suspect that it may represent a fat-specific and structurally distinct complex (Fat-TFIID) because it behaved in a manner distinct from canonical TFIID by size exclusion chromatography (data not shown). These immunoprecipitation assays also revealed that PPARγ co-purifies with TAF7L in the Fat-TFIID complex from differentiated C3H10T1/2 cells but not from control FLAG-V5-GFP cells (*Figure 3A,B*) nor with canonical TFIID lacking TAF7L (data not shown). Although we failed to detect PRDM16 in these affinity purification experiments, we did find that ectopically expressed TAF7L and PRDM16 associate with each other in 293T cells (*Figure 3C*). These data suggest that TAF7L-containing Fat-TFIID has gained the ability to bind endogenous PPARγ, which we speculate might facilitate its association with distinct cofactors such as PRDM16 in BAT or TLE3 in WAT to differentially regulate brown and white adipocyte formation (*Villanueva et al., 2011*, *2013*).

To probe the potential functional relevance of these interactions, we next examined the genome-wide co-occupancy of TAF7L and PPARγ at BAT-selective genes in differentiated C3H101/2 adipocytes. Due to the presence of rosiglitazone in the differentiation regime, our previous ChIP-seq analysis of C3H10T1/2-derived fat cells represented a mixture of white and brown fat cells, as manifested by enhanced *Cidea*, *Elovl3*, and *Ucp1* expression (*Harms and Seale, 2013*). Here, we have re-analyzed the binding site data for TAF7L, TBP, and PPARγ focusing on loci of activated BAT-selective genes (*Cidea*, *Prdm16*, and *Ucp1*) (*Figure 3D*, *Figure 3—figure supplement 1*). As shown in *Figure 3—figure supplement 1A, B*, TAF7L and PPARγ co-occupy overlapping regions at the *Prdm16* and *Ucp1* loci in post-differentiated (-post) cells, but not at control genes (*Ajap1* and *Bmod2*) nor to the same genes in pre-differentiated cells (-pre). These findings suggest that PPARγ and TAF7L likely function as a coupled activator–coactivator pair that regulates BAT-specific gene transcription much as we previously described for WAT differentiation.

To extend our analysis of TAF7L mechanisms in regulating fat-specific gene transcription, we next employed chromatin conformation capture (3C) to assess its participation in transcription factor-mediated long distance DNA looping at two TAF7L-activated genes *Cidea* and *Scd1* (*Figure 3D*, *Figure 3—figure supplement 1*; *Liu et al., 2011*). These two target genes are bound by TAF7L and PPARγ both at their core promoters as well as at their distal enhancers located ~10 kb away. In contrast, TBP only binds at the core promoters of these genes. Our 3C results revealed the likely formation of DNA looping between core promoters and distal enhancers in WT BAT for both loci while significantly reduced looping was seen in BAT lacking TAF7L (*Figure 3D*, *Figure 3—figure supplement 1C*). These data suggest that the presence/incorporation of TAF7L into the putative Fat-TFIID complex likely mediates or enhances long-range chromatin transactions that influence BAT cell fate.

Taken in aggregate, our findings strongly suggest that TAF7L may serve as a key component of an alternative TFIID complex that favors brown fat formation vs muscle lineages. We do not fully understand the molecular mechanisms driving the enhanced formation of skeletal muscle upon loss of TAF7L, but we speculate that TAF7L depletion decreases the transcription of BAT-selective genes, which might indirectly de-repress muscle-selective genes thereby switching precursor cells toward a skeletal muscle lineage. Moreover, the presence of TAF7L potentiates the efficient participation of a specialized Fat-TFIID complex in long-range chromatin interactions. Such a DNA looping mechanism is likely mediated through association with the fat-specific transcription factor PPARγ (*Figure 3E*). These new findings and our previous studies support the notion that TAF7L functions as a common cofactor regulator required for both WAT and BAT development.

## Materials and methods

### Vectors and plasmids

Full-length green fluorescent protein (GFP) or *Taf7l* cDNAs were inserted into p3XFLAG-CMV-10 vector to construct pCMV-FLAG-V5-GFP/TAF7L or into pCS2+ vector to construct pCS2+HA-TAF7L; full-length PRDM16 cDNA was inserted into p3XFLAG-CMV-10 vector to construct pCMV-FLAG-PRDM16.

### Antibodies

TAF4 (612054; BD, San Jose, CA), TBP (62126; abcam, Cambridge, MA), V5 (R960-25; Invitrogen, Carlsbad, CA), HA (9110; abcam), FLAG (F31655; Sigma, Saint Louis, MI) MYHC (05-716; Millipore, Billerica, MA), FABP4 (66682; abcam), UCP1(10983; abcam), TAF7L (prepared in-house, Covance,

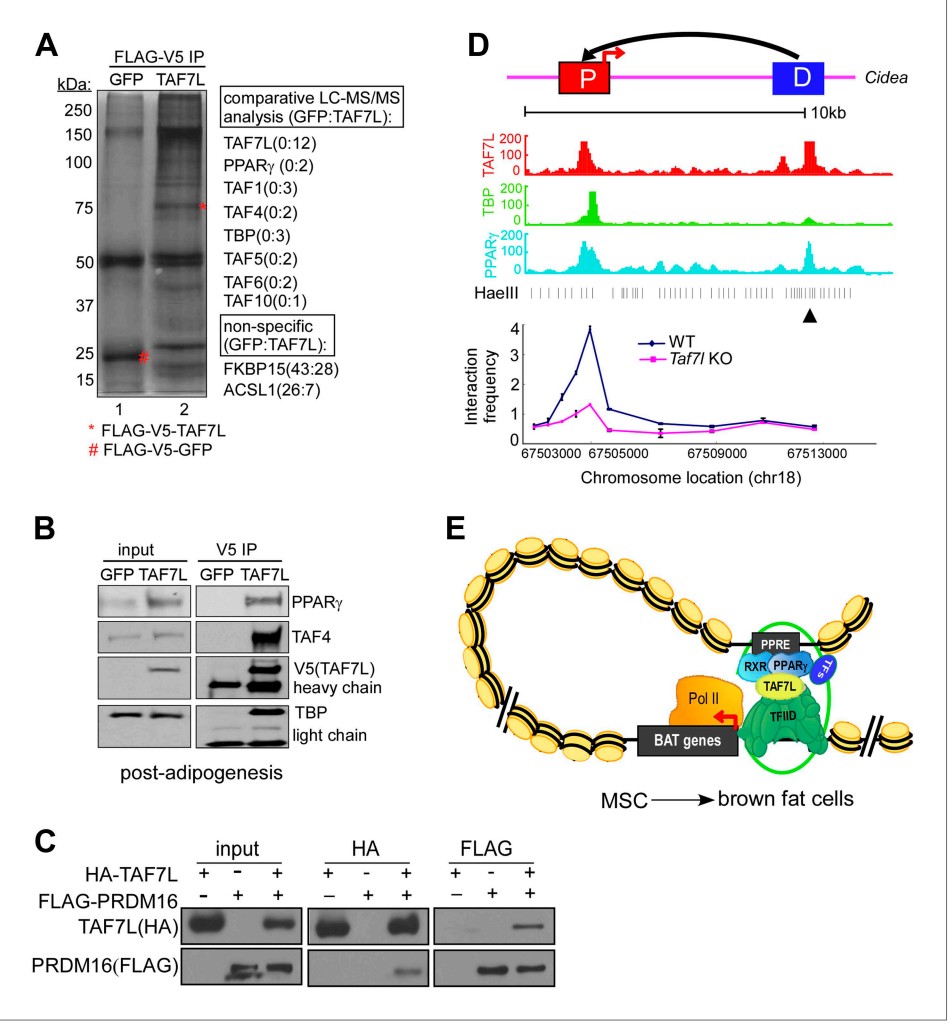

**Figure 3**. TAF7L within a Fat-TFIID associates with PPARγ and facilitates DNA looping formation. (**A**) Silver staining shows co-immunoprecipitated proteins in FLAG-V5-GFP-expressing (GFP, lane 1) and FLAG-V5-TAF7L-expressing (TAF7L, lane 2) C3H10T1/2 differentiated fat cells. Comparative LC-MS/MS analysis identified peptides matching with TFIID subunits (TAF1, TAF4, TBP, TAF5, TAF6, and TAF10) and PPARγ in TAF7L-expressing but not in GFP-expressing cells. FKBP15 and ACSL1 are representative non-specific associated proteins. (**B**) Western blot analyzing input and immunoprecipitated protein levels of PPARγ, TAF4, TAF7L, and TBP in samples from **A**. (**C**) HA-tagged TAF7L and FLAG-tagged PRDM16 were overexpressed in 293T cells, immunoprecipitations were performed on both FLAG and HA antibodies and followed by Western blotting with FLAG and HA antibodies. (**D**) Upper panel, schematic picture shown the distance between distal enhancer (D) and core promoter (P) of *Cidea* gene; middle panel, read accumulation of TAF7L, TBP, and PPARγ on *Cidea* locus in differentiated fat cells from ChIP-seq analysis; bottom panel, 3C experiments assess long-range DNA interactions between the TAF7L/PPARγ binding distal enhancer (D) and core promoter (P) sites of *Cidea* in WT and *Taf7l* KO BAT. ▲, anchor point. Also see ***Figure 3—figure supplement 1C***. (**E**) Model shows TAF7L-mediating regulatory DNA looping to specify BAT differentiation from mesenchymal stem cells (MSC).

The following figure supplements are available for figure 3:

**Figure supplement 1**. TAF7L colocalizes with PPARγ on core promoters and enhancers of BAT-specific genes.

---

Denver, PA), Pol II (monoclonal 8WG16, protein-A purified), PPARγ (sc-7196), ANTI-FLAG M2 Affinity Gel (A2220; Sigma), Anti-V5-agarose affinity gel (A7345; Sigma).

## Cells culture, stable cell line establishment

C3H10T1/2 and C2C12 cells were cultured in high glucose DMEM with 10% fetal bovine serum at 5% $CO_2$.

C3H10T1/2 cells stably expressing FLAG-V5-GFP/TAF7L were established by transfection of the pCMV-FLAG-V5-GFP/TAF7L plasmids followed by 2 weeks of 1 µg/µl neomycin selection.

## Brown adipocyte differentiation, Oil red O staining, and C2C12 myogenesis

For adipogenesis, C3H10T1/2, control and FLAG-TAF7L-expressing C2C12, and FLAG-V5-GFP/TAF7L-expressing C3H10T1/2 cells were grown in high glucose DMEM supplemented with 10% fetal bovine serum. At confluence, cells were exposed to induction medium containing dexamethasone (1 µM) (265005, CalBiochem, San Diego, CA), isobutylmethylxanthine (IBMX, 0.5 mM) (I-5879, Sigma), Indomethacin (0.125 mM) (I-7378, Sigma), and 10% FBS. 3 days later, cells were further cultured in high glucose DMEM-containing insulin (5 µg/ml) (19278, Sigma), T3 (0.1 µM) (T-2877, Sigma), and rosiglitazone (5 µM) (71740, Cayman, Ann Arbor, MI) until they were ready for harvest.

For Oil red O staining, pre- and post-differentiated C3H10T1/2 cells, shGFP and shTAF7L-treated C3H10T1/2 cells, C2C12.CNTL and C2C12.TAF7L cells were washed once in PBS and fixed with freshly prepared 4% formaldehyde in 1 × PBS for 30 min, followed by standard Oil red O staining method described previously (*Zhou et al., 2013a*).

C2C12 myogenesis procedure followed a previous study (*Zhou et al., 2013a*).

## RNA isolation and real-time PCR analysis

Total RNA from cultured cells or mouse tissues was isolated using QIAGEN RNeasy Plus mini columns according to the manufacturer's instructions (Qiagen, Germany). For RT-qPCR analysis, 1 µg total RNA was reverse transcribed using cDNA reverse transcription kit (Invitrogen). PCR reactions using the SYBR Green PCR Master Mix (Applied Biosystems, Grand Island, NY) were performed according to the manufacturer's instruction using an ABI 7300 real time PCR machine (Applied Biosystems). Relative expression of mRNA was determined after normalization to cyclophilin gene. Student's *t* test was used to evaluate statistical significance (*Zhou et al., 2013b*).

## Western blot analysis, immunoprecipitation, and silver staining

Whole cell extracts were prepared from cells by homogenization in lysis buffer containing 50 mM Tris-Cl, pH 8.0, 500 mM NaCl, and 0.1% Triton X-100, 10% glycerol, and 1 mM EDTA, supplemented with protease inhibitor cocktail (Roche, Indianapolis, IN) and phenylmethylsulphonyl fluoride (PMSF). 15 µg of whole-cell lysates were separated by SDS-PAGE and transferred to nitrocellulose membrane. For immunoblotting, membranes were blocked in 10% milk, 0.1% Tween-20 in TBS for 30 min, and then incubated with TAF7L, UCP1, PPARγ, and POL II antibodies for 2 hr at room temperature; detailed Western blotting procedure was performed as previously described (*Zhou et al., 2006*).

3 mg whole-cell extracts from FLAG-V5-GFP/TAF7L post-differentiated adipocytes were immunoprecipitated with 100 µl of ANTI-FLAG M2 Affinity Gel under the conditions of 0.3 M NaCl, 0.2% NP-40 at 4°C overnight. After extensive washing with buffer containing 0.3 M NaCl and 0.1% NP-40, proteins were eluted from the affinity gel with 100 µg/ml FLAG peptide in 0.1 M NaCl Tris buffer. Elutions from both samples were subsequently immunoprecipitated with 40 µl Anti-V5-agarose affinity gel antibody with a similar procedure as above. After extensive washes, proteins were eluted with 40 µl pH2.5 0.1 M glycine for three times and immediately neutralized with 12 µl of pH9.0 2 M Tris-Cl. 10 µl of eluted samples were subjected to 10% SDS-PAGE and followed by standard silver staining or by western blotting analysis with V5, PPARγ, TAF4, and TBP antibodies to detect tagged-proteins in the inputs and associated proteins as previously described (*Harms and Seale, 2013*). The remaining samples were precipitated by 20% trichloroacetic acid, and the precipitates were sent to liquid chromatography-tandem mass spectrometry (LC-MS/MS) to detect peptides derived from proteins pulled-down by FLAG-V5-GFP/TAF7L.

500 µg whole-cell extracts from 293T cells transfected with HA-TAF7L and FLAG-PRDM16 were immunoprecipitated with FLAG or HA antibodies at 4°C for overnight under the conditions of 0.3 M NaCl and 0.2% NP-40, 30 µl protein A/G beads were added and incubated for additional 2 hr at 4°C, after extensive washing with buffer containing 0.3 M NaCl and 0.1% NP-40, remaining beads were subjected to 10% SDS-PAGE and followed by western blotting analysis with FLAG and HA antibodies to detect tagged-proteins in the inputs and IPs as described previously (*Zhou et al., 2013a*).

## Animals and genotype analysis

The derivation of *Taf7l* KO mice has been previously described (*Kajimura et al., 2009*). All animal experiments were performed in strict accordance with the recommendations in the Guide for the Care

and Use of Laboratory Animals of the National Institutes of Health. All of the animals were handled according to animal use protocols (#R007) approved by the Institutional Animal Care and Use Committee (IACUC) of the University of California, Berkeley. Mice were maintained on a standard rodent chow diet with 12 hr light and dark cycles. *Taf7l* KO mouse line was maintained on a C57BL/6J background. Genotyping was performed by PCR as previously described (*Cheng et al., 2007*).

### Immunohistochemistry

For histological analysis on interscapular BAT of E18.5 embryos from WT and *Taf7l* KO mice, freshly-harvested mouse embryos were genotyped and the interscapular regions of embryos were transversally dissected and then fixed in 10% formaldehyde for 24 hr at 4°C; tissue was embedded in paraffin using the microwave method as described and then sectioned into 8–10 µm to mount on slides (*Zhou et al., 2013b*). The following immunohistochemistry by haematoxylin and eosin (H&E) staining and FABP4, UCP1, and MYHC immunostaining were performed using the method described previously (*Zhou et al., 2013a*).

### Preparation of primary brown adipocytes and brown fat differentiation

Fresh interscapular brown adipose tissues were removed from 3-week-old euthanized WT and *Taf7l* KO mice and finely minced, digested with 0.25% trypsin for 30 min at 37°C, and centrifuged for 5 min at 2,000×*g* to get rid of debris. The pellet was resuspended in culture media before plated on gelatin coated plates. Cells were cultured at 37°C in high glucose DMEM supplemented with 20% FBS. Brown adipocyte differentiation and staining were followed the same procedure as C3H10T1/2 cells.

### mRNA-seq libraries preparation and deep sequencing

Total RNA was extracted from BAT, carefully excised to get rid of surrounding tissue based on the texture and color, of 6 WT and 6 *Taf7l* KO mice. RNA was extracted separately for each mouse by RNeasy Plus Mini Kit (Qiagen) and then pooled for WT or *Taf7l* KO samples; 4 µg of each RNA pool was used to purify mRNA using oligo (dT) and subsequently converted into multiplexed mRNA-seq libraries using mRNA-Seq Trueseq Kit (Illumina, San Diego, CA). Samples were multiplexed and sequenced in one lane on an Illumina HiSeq 2000 sequencer (QB3 Vincent J Coates Genomics Sequencing Library, University of California, Berkeley, CA). 50 bp single-end reads were used for both samples; each sample produced over 30 million reads.

### Digital gene expression of mRNA-seq and gene ontology analysis

Reads were mapped to the mouse transcriptome (mm10), using TopHat (*Langmead et al., 2009*; *Trapnell et al., 2009*), version v1.4.0., with default parameters. We then applied cufflinks (*Trapnell et al., 2010*), version v1.3.0, using the default parameters except: --max-mle-iterations 1, to estimate the digital expression levels at each transcript. Gene ontology analysis was done using DAVID Bioinformatics Resources 6.7.

### Chromosome conformation capture (3C)

3C analysis was performed as previously described on WT and *Taf7l* KO BAT (*Liu et al., 2011*), which was carefully excised to get rid of all other possible tissue based on the brown color and the tissue texture. 2 mg of freshly dissected interscapular BAT from 6 WT or 6 *Taf7l* KO mice were minced, homogenized extensively to nearly single cells and washed, crosslinked with 1% formaldehyde for 15 min at 4°C and then quenched with 0.125 M glycine for 5 min.

Crosslinked BATs were lysed and the chromatin was digested with 8 U HaeIII (NEB) for the *Cidea* and *Scd1* loci. Digested fragments were cleaned and subsequently ligated with 60 units T4 DNA ligase (Invitrogen) for 4 hr at 16°C. Following proteinase K digestion and decrosslinking at 65°C overnight, DNA fragments was recovered by phenol–chloroform extraction.

Control templates were generated using individual BAC clones covering *Cidea* or *Scd1* locus (Bacpac). 10 µg of BAC DNA was digested with 20 units HaeIII and then randomly ligated with 10 units T4 DNA ligase in 50 µl volume.

3C primers were designed upstream and downstream of the core promoter site (P). Primers annealing to distal enhancers (D) corresponding to TAF7L and PPARγ binding sites on either *Cidea* or *Scd1* were used as anchor points. 3C analysis was done by qPCR using as a primer pair the anchor point primer and one annealing to region under investigation. Each data point in WT and *Taf7l* KO BAT was normalized by the BAC control template and presented as interaction frequency.

## Data availability

Raw and mapped sequencing reads are available from the National Center for Biotechnology Information's GEO database (http://www.ncbi.nlm.nih.gov/geo/) under accession number GSE55797. Primer sequences are listed in *Supplementary file 1*.

## Acknowledgements

We thank PJ Wang for providing *Taf7l* KO mice; S Kajimura for protocols and suggestions; R Steven and D Schichness for help with immunochemistry; G Dailey, S Zheng, M Haggart, C Cattoglio, SS Teves, SE Torigoe, and FL Xie for their kind help and useful discussions. H Zhou is a research associate of the Howard Hughes Medical Institute. T Kaplan is a member of the Israeli Center of Excellence (I-CORE) for Gene Regulation in Complex Human Diseases (no. 41/11), and the Israeli Center of Excellence (I-CORE) for Chromatin and RNA in Gene Regulation (1796/12). R Tjian is an investigator of the Howard Hughes Medical Institutes.

## Additional information

### Competing interests

RT: Robert Tjian is President of the Howard Hughes Medical Institute (2009-present), one of the three founding funders of *eLife*. The other authors declare that no competing interests exist.

### Funding

| Funder | Author |
| --- | --- |
| Howard Hughes Medical Institute (HHMI) | Haiying Zhou, Ivan Grubisic, Robert Tjian |

The funders had no role in study design, data collection and interpretation, or the decision to submit the work for publication.

### Author contributions

HZ, Conception and design, Acquisition of data, Analysis and interpretation of data, Drafting or revising the article, Contributed unpublished essential data or reagents; BW, RT, Conception and design, Analysis and interpretation of data, Drafting or revising the article; IG, TK, Acquisition of data, Analysis and interpretation of data, Drafting or revising the article

### Ethics

Animal experimentation: All animal experiments were performed in strict accordance with the recommendations in the Guide for the Care and Use of Laboratory Animals of the National Institutes of Health. All of the animals were handled according to animal use protocols (#R007) approved by the Institutional Animal Care and Use Committee (IACUC) of the University of California, Berkeley.

## Additional files

### Supplementary file

• Supplementary file 1. Primer sequences for RT-qPCR experiments (upper panel) and 3C experiments (lower panels).

### Major datasets

The following dataset was generated:

| Author(s) | Year | Dataset title | Dataset ID and/or URL | Database, license, and accessibility information |
| --- | --- | --- | --- | --- |
| Zhou H, Wan B, Grubisic I, Kaplan T, Tjian R | 2014 | Taf7l Modulates Brown Adipose Tissue Formation | http://www.ncbi.nlm.nih.gov/geo/query/acc.cgi?acc=GSE55797 | Publicly available at NCBI Gene Expression Omnibus. |

**Reporting standards:** Standard used to collect data: The data was followed NCBI standards for uploading mega-data set.

The following previously published dataset was used:

| Author(s) | Year | Dataset title | Dataset ID and/or URL | Database, license, and accessibility information |
|---|---|---|---|---|
| Zhou H, Kaplan T, Li Y, Grubisic I, Zhang Z, Wang PJ, Eisen MB, Tjian R | 2012 | Dual Functions of TAF7L in Adipocyte Differentiation | http://www.ncbi.nlm.nih.gov/geo/query/acc.cgi?acc=GSE41937 | Publicly available at NCBI Gene Expression Omnibus. |

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
