## [Decision Letter]

Thank you for sending your work entitled “TAF7L Modulates Brown Adipose Tissue Formation” for consideration at *eLife*. Your article has been favorably evaluated by Fiona Watt (Senior editor) and 3 reviewers, one of whom, Jim Kadonaga, is a member of our Board of Reviewing Editors.

The Reviewing editor and the other reviewers discussed their comments before we reached this decision, and the Reviewing editor has assembled the following comments to help you prepare a revised submission.

The manuscript “TAF7L Modulates Brown Adipose Tissue Formation” by Zhou et al. provides new evidence for the role of TAF7L in brown fat development. Brown and white fat are now appreciated to come from different developmental lineages, and the transcriptional control of their development is an area of active research. The work presented in this paper suggests an important role for a new molecular player, TAF7L. While the data shown are quite exciting, additional experiments, particularly BAT-specific studies, should be performed to support the conclusions of this manuscript.

Major points:

1) In Figure 3, the authors used 10T1/2 cells to study the mechanism whereby TAF7L modulates brown fat development. TAF7L was overexpressed, and the complex containing this protein was purified and shown to contain PPARg. In a previous paper, the authors had observed that these two proteins interact in 293T cells. Since PPARg is required for white and brown adipocyte development, it is not evident how this interaction specifically modulates brown adipogenesis. To address this point, the authors performed ChIP-Seq and found that TAF7L and PPARg co-occupy overlapping regions of PRDM16 and Ucp1 in post-differentiated cells. To clarify this line of investigation, the authors should address the following questions.

a) What other proteins were in the TAF7L complex? Were there any brown-fat enriched proteins that TAF7L interacts with such as PRDM16, PGC1a, or EHMT1?

b) Might TAF7L modulate PPARg selectivity for PRDM16 (and brown fat) vs. TLE3 (and white fat)?

2) In their previous paper (Zhou et al., *eLife*, 2013), the authors showed that white fat development (in vitro and in vivo) is disrupted in the absence of TAF7L. This finding argues for a broader role of TAF7L in adipogenesis (or perhaps even more broadly in development). As highlighted in point 1, the authors should address how TAF7L is specifically involved in brown fat development. In other words, the role of TAF7L in the browning of WAT should be addressed.

3) Photographs of whole BAT from young and adult WT and TAF7L KO mice should be provided to see the gross morphology of BAT.

4) Cold exposure experiments should be performed in TAF7L KO mice to understand the physiological consequence of lack of TAF7L in BAT.

5) The mechanism of how TAF7L suppresses muscle-specific genes is unclear. This point should be further investigated or minimally speculated on and discussed.

6) Differentiation assays using primary brown adipocytes (SVF differentiated to adipocytes) from WT and TAF7L KO mice should be performed.

---

## [Author Response]

We thank the reviewers for a thoughtful and comprehensive review of our manuscript “TAF7L Modulates Brown Adipose Tissue Formation”, which we here, resubmit as a Short report to *eLife*. As a significant advancement of our original finding that TAF7L ablation disrupts white fat tissue development (Zhou et al., *eLife*, 2013), we report that TAF7L forms a key component of an alternative Fat-TFIID complex that works in conjunction with PPARγ via a DNA looping mechanism to modulate brown adipose tissue (BAT) formation.

A note from the editors appended below outlines the critical changes to our manuscript that were recommended by the BRE and reviewers:

*- Two points (Major Points 1a, 3) involve data that the authors probably already have*.

*- Several issues (Major Points 1b, 2, 5) could be addressed simply by discussing them in the text*.

*- Major Points 4 and 6 would make it a better paper, but it could be argued that such experiments are beyond the scope of this study. It's up to the authors whether they want to strengthen this work, particularly with regard to brown fat development and physiology*.

We appreciate these many useful suggestions from reviewers, which has further strengthened and broadened this study. In response to the reviewers’ comments, we have now included new data to address major points 1a, 3, and 6. Below, we also address points 1b, 2, 4 and 5, which includes suggestions that we believe are beyond the scope of this study.

*Major*
*points:*

*1) In*
Figure 3*, the authors used 10T1/2 cells to study the mechanism whereby TAF7L modulates brown fat development. TAF7L was overexpressed, and the complex containing this protein was purified and shown to contain PPARg. In a previous paper, the authors had observed that these two proteins interact in 293T cells. Since PPARg is required for white and brown adipocyte development, it is not evident how this interaction specifically modulates brown adipogenesis. To address this point, the authors performed ChIP-Seq and found that TAF7L and PPARg co-occupy overlapping regions of PRDM16 and Ucp1 in post-differentiated cells. To clarify this line of investigation, the authors should address the following questions*.

*a) What other proteins were in the TAF7L complex? Were there any brown-fat enriched proteins*
*that TAF7L interacts with such as PRDM16, PGC1a, or EHMT1?*

Although we made multiple attempts, we were unable to recover sufficient levels of protein from the IP experiments using adipocytes to perform unambiguous protein identification. However, we did find that TAF7L can interact with PRDM16 and be coIP'd when they are expressed in 293T cells. We have now included this new data in Figure 3.

*b) Might TAF7L modulate PPARg selectivity for*
*PRDM16 (and brown fat) vs. TLE3 (and white fat)?*

We have not tested the association between TLE3 with TAF7L in WAT, but we did find that TAF7L can associate with PRDM16 when ectopically expressed in 293T cells. We have now revised the text to read: Although we failed to detect PRDM16 in these affinity purification experiments, we did find that ectopically expressed TAF7L and PRDM16 associate with each other in 293T cells (Figure 3). These data suggest that TAF7L-containing Fat-TFIID has gained the ability to bind endogenous PPARγ we speculate which might facilitate its association with distinct cofactors such as PRDM16 in BAT or TLE3 in WAT to differentially regulate brown and white adipocyte formation.

*2) In their previous paper (Zhou et al.,* eLife, *2013), the authors showed that white fat development (in vitro and in vivo) is disrupted in the absence of TAF7L. This finding argues for a broader role of TAF7L in adipogenesis (or perhaps even more broadly in development). As highlighted in point 1, the authors should address how TAF7L is specifically involved in brown fat development. In other words, the role of TAF7L in the browning of WAT should be addressed.*

We agree that this is a reasonable scenario since *Taf7l* KO mice showed changes consistent with the browning of WAT and changes in the expression of WAT genes in *Taf7l* KO mice compared to WT mice. However, we believe a thorough and rigorous test of this hypothesis is well beyond the scope of this current study. We are particularly concerned that fully dissecting the browning of WAT will require a careful analysis of beige cells which are likely not the same as canonical BAT.

*3) Photographs of whole BAT from young and adult WT and TAF7L KO mice should be provided to see the gross morphology of BAT*.

We have now included photographs of whole BAT from one-month and four-month old WT and TAF7L KO mice in Figure 1—figure supplement 1.

*4) Cold exposure experiments should be performed in TAF7L KO mice to understand the physiological consequence of lack of TAF7L in BAT*.

In this Short report we chose to focus primarily on the role of TAF7L with respect to BAT developmental. In addition, our preliminary physiological studies showed mixed phenotypes because TAF7L KO resulted in differential changes in WAT versus BAT development. We felt that it would be least ambiguous to study the physiological consequences of environmental parameters such as cold exposure using BAT or WAT-specific TAF7L KO mice in future studies.

*5) The mechanism of how TAF7L suppresses muscle-specific genes is unclear. This point should be further investigated or minimally speculated on and discussed*.

Although the mechanisms of how TAF7L suppresses muscle-specific genes is unclear at this point, we speculate that it might be an indirect effect, since our ChIP-seq data showed that TAF7L is not targeted to muscle gene promoters. We therefore favor the idea that a de-repression of muscle-genes might result from a reduction in certain fat genes caused by loss of TAF7L, which we establish here to be functioning as an activator of fat cell lineages. We have now included this speculation in the revised text.

*6) Differentiation assays using primary brown adipocytes (SVF differentiated to adipocytes) from WT and TAF7L KO mice should be performed*.

We have now included this new data in a revised Figure 2—figure supplement 1.